# Carbon Emissions and Firm Performance: Evidence from Financial and Non-Financial Firms from Selected Emerging Economies

**Mohammad Dulal Miah** [1,*]**, Rashedul Hasan** [2] **and Mohammed Usman** [3]

1   Department of Economics and Finance, University of Nizwa, Nizwa 616, Oman
2   Department of Economics, Finance, and Accounting, Coventry University, Coventry CV1 5ED, UK; ad6284@coventry.ac.uk
3   Department of Management, University of Nizwa, Nizwa 616, Oman; usman@unizwa.edu.om
*   Correspondence: dulal@unizwa.edu.om

**Abstract:** This paper examines the effects of carbon emissions on the accounting and market-based performance of financial and non-financial firms in emerging economies. Data for 104 financial and 328 non-financial firms constituting 2591 observations operating in 22 emerging economies were collected from the Datastream database for the period 2011–2020. We applied OLS and 2SLS regression techniques to analyze the data. Results show that financial firms emit less carbon than their non-financial counterparts. The results further show that carbon emissions reduce firms' return on equity, Tobin's Q, Z-score, and credit rating. Our findings remain robust in different estimation techniques and alternative proxies of performance. Our results have some important policy implications for emerging economies.

**Keywords:** environmental performance; carbon emission; emerging economies; firm performance

## 1. Introduction

This research aims to investigate the effect of firms' carbon emissions on performance for a large sample of financial and non-financial firms in emerging economies. More specifically, the research examines if firms' direct and indirect carbon emissions affect the return on assets, Tobin's' Q, Z-score, and credit rating, and if the results differ between financial and non-financial firms across regions. Further, this research attempts to examine if the effects of firms' carbon emissions vary between state-owned, rent seeking, and environment project financing firms.

Although developed countries are mostly responsible for the mounting carbon stock in the atmosphere, emerging countries are gradually evolving as leading emitters. Recently, emerging economies including Brazil, Russia, India, and China (BRICs) have experienced a rapid rise in GDP growth rate resulting from higher productivity as well as increased economic activities, which require a huge energy input to keep the nascent pace of economic growth stable. On the other hand, structural changes in economic activities accompanied by the augmented GDP growth rate have resulted in rapid urbanization. These effects combined lead to higher carbon emissions in the atmosphere [1–3]. A report published by the International Energy Agency [4] shows that the decline in emissions in advanced economies accounted for 10% in 2020 compared to 2019, mainly due to the disruptive economic activities caused by the COVID-19 pandemic. However, emissions from emerging economies declined on average only by 4% during the same period. China, the largest carbon emitter at present, experienced an increase in emissions in 2020 and in the first two quarters of 2021 [5,6]. This implies that achieving the target level of emission cut depends decisively on carbon reduction strategies in emerging countries.

Despite widespread importance, the impact of carbon emissions on the accounting and market-based performance of firms in emerging economies and the varying impact on

financial and non-financial firms remain largely unexplored. Our paper aims to fill this important research gap. It is argued that carbon-intensive firms suffer financially due to various reasons. For example, carbon emissions increase firms' cost of capital, regulatory compliance, and litigation costs. Moreover, firms with a high carbon profile are required to pay a larger carbon premium under the ETS. On the other hand, customers hesitate to use products and services that are not environmentally friendly [7]. Furthermore, carbon emissions ruin firms' reputation, which results in a lower financial and market-based performance of high-carbon-emitting firms.

To achieve the above stated objective, we collected data for 104 financial and 328 non-financial firms from 22 emerging countries. After winsorizing 1% at both tails, our final dataset yielded 2591 observations for the period 2011–2020. We applied ordinary least squares (OLS) and two-stage least squares (2SLS) regression techniques to analyze the data. The results show that financial firms emit less carbon than their non-financial counterparts. The results further show that firms' carbon emissions reduce returns on equity, Tobin's Q, Z-score, and credit rating. However, the degree of performance decline attributed to carbon emission is more pronounced for non-financial firms than financial firms.

Our research contributes to the current literature in some important ways. First, although the impact of firms' carbon emissions on financial and market performance is well explored in the context of advanced economies as well as in single-country contexts, this issue remains largely unexplored in the case of emerging economies. Moreover, our dependent variables are direct and indirect carbon emissions, which provide a better view than the aggregate carbon emission data used in most studies because firms may put efforts into cutting direct emissions while remaining unaware of the effects of indirect carbon emissions. Hence, our results provide an important insight for managers as to the effects of direct and indirect carbon emissions on firms' financial and market-based performance. In particular, the finding of the research is believed to provide managers with economic motivations to be environment friendly.

Second, unlike some studies that either exclude financial firms or consider them together with the non-financial corporations, we treat them separately and combined to derive a better view and examine if the effect of carbon emissions on performance varies between these two different clusters of firms. Financial firms operate in a different regulatory environment compared to their non-financial peers. Moreover, it is a common belief that financial firms are comparatively clean and less damaging to environment. However, financial firms can play a pivotal role in mitigating the carbon footprint across the globe and can benefit from reducing their emissions through implementing green financing codes in their operations as well as creating incentive and sanction mechanisms for client firms. The findings of our research would provide evidence if financial and non-financial firms can benefit from voluntary mitigation of emissions.

Third, our findings provide practical tools for policymakers to restrict firms' carbon footprints through enacting various rules. In other words, our finding that carbon emissions reduce firms' financial performance and market value provides an economic rationale for policymakers to enact new regulations for mitigating firm-level carbon emissions. Additionally, our research contributes to the field not only by providing new evidence on the relationship between firms' carbon emissions and financial as well as market performance but also by advancing the debate as to what role emerging economies can play to enhance the effort toward a greener planet. The world is supposed to witness a net decline in carbon emissions, but the total anthropogenic emissions are on course to rise again after a short break in 2020 owing to the lower economic activities globally due to the COVID-19 pandemic. The results of this study thus provide an economic rationale for stakeholders to be environment friendly.

The rest of the paper is structured as follows: Section 2 discusses the relevant literature that helped us develop the hypotheses. Section 3 explains the data and methodology, whereas Section 4 presents and interprets the results. Section 5 concludes the paper

by summarizing the significant findings, offering some policy recommendations, and identifying future research areas.

## 2. Literature Review and Hypothesis Development

Prior literature investigates the impact of carbon emissions on various attributes of firms [8–10]. For instance, Jung et al. [11], Kim et al. [9], and Li et al. [12] examined the impact of firms' carbon emissions on the cost of bank debt financing for Australian firms and found a positive association between them. In a similar fashion, Du et al. [13] for Chinese private-owned firms, El Ghoul et al. [14] for a sample of manufacturing firms in 30 countries, Kleimeier and Viehs [10] drawing evidence from 58 countries, and Nandy and Lodh [15] for US firms found consistent evidence that higher carbon emissions have a positive and significant effect on the cost of bank debt financing. Similarly, Chapple et al. [16] for a sample of 58 Australian listed firms, Griffin et al. [17] and Matsumura et al. [18] for S&P 500 firms, Johnston et al. [19] in the context of US electric firms, Lee et al. [20] using a sample of 362 firms, and Wen et al. [6] for Chinese corporate entities show that firms' carbon emissions reduce equity value. Kabir et al. [8] provide evidence that carbon emissions increase firms' credit risk, which, in turn, lowers the credit rating of carbon-intensive firms [21].

Firms' carbon reduction also helps boost their financial return. Former vice president of the United States Al Gore [22] (p. 342) stated "3M, in its Pollution Prevention Pays program, has reported significant profit improvement as a direct result of its increased attention to shutting off all the causes of pollution it could find." This example clearly hints that firms can benefit from carbon mitigation strategies. The literature also provides supporting evidence. Busch and Hoffmann [23] examine the relationship between carbon mitigation and performance of the largest companies of the Dow Jones Global Index and report a positive association between them. Ganda and Milondzo [24] using a sample of 63 South African firms and Busch and Lewandowski [25] through a metadata analysis of 34 studies show a negative impact of carbon emissions on firms' financial performance. Similarly, Cucchiella et al. [26] found that appropriate control of emissions led to a higher profit for Italian firms by increasing demand and productivity. Dechezleprêtre et al. [27] show that the European Union Emission Trading Scheme (ETS) has helped reduce firms' carbon emissions, which, in turn, has led to an increase in firms' revenues. Lewandowski [28] illustrates that performance improvement is pronounced for firms with superior carbon performance but not for companies with inferior carbon performance. Alvarez [29] found evidence of a positive impact of carbon emissions on ROA but not on ROE.

The impact of firms' carbon emissions on performance has been argued from two interrelated perspectives. The first line of argument in the literature argues that firms' carbon emissions involve increased costs of compliance, litigation, and cleaning up, etc. which results in a decline in performance. Bauer and Hann [30] show that litigation risk is higher for firms that fail to mitigate emissions. Companies with a clean environmental profile are likely to face fewer regulatory burdens because of the lower probability of being punished for environmental delinquency. Palmer, Oates, and Portney [31] argue that costs imposed by some environment regulations can be so high that firms may face bankruptcy. Similarly, Chan, Li, and Zhang [32] report that the increased cost of complying with the European Union Emissions Trading System accounts for five to eight percent of the total material cost of carbon-intensive firms. Similarly, Pasurka [33], analyzing nine countries in Europe, North America, and Asia, reports that pollution abatement costs in 2000 ranged from one percent in Taiwan to five percent in Canada.

Increased production costs imposed by environmental regulations may result in a loss of firms' national and international competitiveness, primarily if countries differ in terms of stringency of environmental regulations. Porter and Linde [34] and Huang, Zhao, and Cao [35] provide evidence that stringent enforcement of environmental regulations accelerates the innovation of low-carbon technologies that require comparatively less energy inputs per unit, which, in turn, helps firms gain production efficiency. Moreover,

firms with a low carbon profile are flexible and well prepared to adopt any regulatory changes that aim to restrict their carbon footprint.

Emissions are a negative externality. Hence, high-carbon-emitting firms are increasingly liable for internalizing negative externalities. Firms that fail to sufficiently commit to this social requirement face adverse social consequences. Frooman [36] argues that stakeholders possess a resource- 'withholding' capacity. For example, suppliers of funds can restrict the flow of finance, whereas regulatory authorities can deny the license or permit for an entity that fails to comply with certain environmental regulations. These events may require high-carbon firms to commit resources for compliance, environmental disaster management, and litigation settlement [37]. Ferris and McGartland [38] report that high-emitting industries including the pulp and paper, oil refinery, and steelmaking industries in the USA incurred regulatory compliance costs equivalent to one percent of their annual turnover in 2005, whereas the corresponding figure for all manufacturing plants was only 0.4 percent.

While the above stream of literature emphasizes on the additional costs incurred by carbon-intensive firms, the second line of argument regards the view that firms' carbon emissions reduce cash flows. Reputation and brand image positively impact firms' sales [39]. Owing to extensive media coverage and increased public awareness, firms' carbon profiles are known to a wider group of stakeholders who can significantly affect firms' reputation and revenue. For example, firms that fail to furnish sufficient tools to control emission and are reluctant to put adequate measures to decarbonizing the environment are not well perceived by stakeholders [20,40,41]. Society punishes such firms by withdrawing support, which may result in serious setbacks including a decline in sales, funding opportunities, and market competitiveness. Sanjuán et al. [42] in the context of Spanish buyers, and Sakagami et al. [43] analyzing the buying behavior of Japanese consumers show that customers are willing to pay a premium for green products. Furthermore, reputational risk also leads partner firms to severe business ties with the polluters. This could seriously disrupt the existing supply chain, leading to a substantial financial loss.

Compliance costs and reputational risks increase the credit risk of carbon-intensive firms by creating contingent liabilities. Credit rating agencies thus incorporate environmental elements in assessing firms' credit risk [21,44,45]. For example, S&P [46] reports that between 2015 and 2017, environmental and climate concerns affected corporate ratings in 717 cases, 10 percent of corporate rating assessments. Similarly, Thompson and Cowton [47] document that 60 percent of banks in the UK have incorporated environmental dimensions in their formal corporate lending policy. Since the debt market holds a significant share of corporate finance, it is highly likely that creditors provide benefit to low-emitting firms by charging low-carbon premiums and punishing high-carbon emitting firms by asking for higher default premiums. This indicates that firms with high carbon emissions, inter alia, suffer financially compared to their low-emitting counterparts. The above discussion leads to the following hypothesis

**Hypothesis 1 (H1).** *Firms' carbon emissions are negatively related to financial and market performance.*

## 3. Data and Methodology

Our study focuses on the effects of carbon emissions on the performance of financial and non-financial firms in emerging economies. We follow the International Monetary Fund's [48] definition of emerging economies in their latest publication of the World Economic Outlook, 2021. We cover a total of 432 firms (104 financial and 328 non-financial firms) operating in six continents, covering ten years starting from 2011. Our dataset is an unbalanced panel consisting of 2591 observations. We collected carbon emission and firm-specific information for the selected firms from Datastream and macroeconomic data from the World Bank database. Table A1 in the Appendix A provides the distribution of our sample.

To explore the differences in the consequences of carbon emissions between financial and non-financial firms, we ran a regression using the following equation:

$$Carbon\ emission_{it} = \beta_0 + B_1 Financial\ dummy_{it} + B_2 Controls_{it} + \varepsilon_{it} \qquad (1)$$

$$Financial\ perfomance_{it} = \beta_0 + B_1 Carbon\ emission_{it} + B_2 Controls_{it} + \varepsilon_{it} \qquad (2)$$

In Equation (1), carbon emission is the dependent variable, and we used total carbon emission as a proxy for the baseline model. Afterwards, we introduced both direct and indirect carbon emissions as robust proxies for carbon emission. The financial dummy was a dichotomous variable, taking the value of 1 for financial firms and 0 for non-financial firms. In Equation (2), we explored the impact of carbon emissions on financial performance. We used both accounting and market measures of firm performance. Return on assets and earnings per share were our proxies for accounting performance. Tobin's Q was the measure of market performance. In addition, we introduced credit score and Z-score to explore the stability of the selected firms. Our empirical models have the same firm- and country-level controls. Firm-level controls included firm size, age, leverage, capital expenditure, board size, independent members on the board, and strategic ownership. Country-level controls included GDP growth and inflation. Brief descriptions for all variables included in the regression model are available in Table A2 (Appendix A).

## 4. Empirical Results and Discussion

### 4.1. Descriptive Statistics

We present the descriptive statistics in Table 1. Panel A reports the performance variables, and we used return on assets, earnings per share, Tobin's Q, credit score, and Z-score as proxies for performance measures. Return on assets had a mean score of 5.2 percent with a standard deviation of 0.067. We found that the return on assets score ranged between −13.5 percent and 30.6 percent, indicating a diverse pool of financial and non-financial firms in terms of accounting performance. Earnings per share also showed similar statistics, with a mean score of 4.1 percent and a standard deviation of 0.172. Tobin's Q was the proxy of firm value in our study, and we report a mean Tobin's Q of 1.713. Such a score indicates that, on average, the market value of our selected firms is higher than the book value of assets. Additionally, we explored the stability of financial and non-financial firms using credit score and Z-score proxies. The mean credit score was 9.07 with a standard deviation of 3.07. The credit score statistics become clear by looking at the minimum and maximum values, which were 1 and 17, respectively. A higher credit score indicates a good credit rating and vice versa. The mean Z-score was 3.19 with a standard deviation of 3.18.

This study applied three proxies, i.e., total, direct, and indirect carbon emissions. While total carbon emission was the proxy for all baseline equations, we introduced direct and indirect carbon emission as proxies to test the robustness of our results. Total carbon emissions had a mean score of 5.51 with a standard deviation of 1.04. The financial dummy was the explanatory variable in this study as our primary aim was to compare the consequences of carbon emissions between financial and non-financial firms. As such, the minimum and maximum scores for financial dummy ranged between 0 and 1, 1 for financial firms and 0 otherwise. The mean of the financial dummy was 0.20 with a standard deviation 0.40.

We also introduced both firm- and country-level controls in our study. Firm-level controls included firm size, age, leverage, capital expenditure, board size, independent members on the board, and strategic ownership. We had a mixture of firms in terms of size (mean = 9.94) and age (mean = 3.39). On average, our selected firms were highly levered. Debt accounted for 3.13 times the equity. Our sample firms spent a good amount for the acquisition of physical assets (mean = 8.32). We found that, on average, our selected firms had 12 members on their board, and 46.85 percent of their board members were independent members, which is considerably high. The maximum (84.61) and minimum (9.09) numbers of independent board members indicates that there is a large

variation among firms in terms of recruiting independent members on their boards. Only 19.67 percent of the sample firms had strategic ownership (the number of shares held by strategic investors such as corporations, holding companies, individuals, and government agencies). GDP growth and inflation rates were two country-level controls in our empirical model. We found that average GDP growth for our sample countries was 3.17 percent with an inflation rate of 4.75 percent. We checked multi-collinearity issues using correlation analysis and present the results in Table A3 in the Appendix. We did not find a high correlation among independent variables. Hence, the multi-collinearity problem did not bias our results.

**Table 1.** Descriptive statistics.

| Variables | Obs. | Mean | Std. Dev. | Min | Max |
|---|---|---|---|---|---|
| Panel A: Performance variables | | | | | |
| Return on assets | 2575 | 0.052 | 0.067 | −0.135 | 0.306 |
| Earnings per share | 2533 | 0.041 | 0.172 | −1.212 | 0.309 |
| Tobin's Q | 1912 | 1.713 | 1.388 | 0.600 | 9.445 |
| Credit score | 2565 | 9.069 | 3.068 | 1.000 | 17.000 |
| Z-score | 2084 | 3.193 | 3.181 | 0.162 | 19.628 |
| Panel B: Carbon emission proxies | | | | | |
| Total carbon emission | 2591 | 5.508 | 1.044 | 3.180 | 7.838 |
| Direct carbon emission | 2591 | 4.857 | 1.454 | 1.602 | 7.820 |
| Indirect carbon emission | 2591 | 4.986 | 0.901 | 2.627 | 6.955 |
| Panel C: Explanatory variable | | | | | |
| Financial dummy | 2591 | 0.202 | 0.401 | 0.000 | 1.000 |
| Panel D: Firm controls | | | | | |
| Firm size | 2026 | 9.942 | 0.687 | 8.619 | 11.973 |
| Firm age | 2468 | 3.397 | 0.755 | 1.099 | 4.804 |
| Leverage | 2588 | 3.132 | 3.535 | 0.109 | 17.923 |
| Capital expenditure | 2044 | 8.325 | 0.724 | 6.193 | 10.030 |
| Board size | 2590 | 11.761 | 3.613 | 6.000 | 22.000 |
| Independent members on board | 2510 | 46.847 | 17.533 | 9.091 | 84.615 |
| Strategic ownership | 2455 | 19.675 | 2.522 | 11.071 | 24.172 |
| Panel E: Country controls | | | | | |
| GDP growth | 2591 | 3.169 | 2.756 | −3.546 | 8.486 |
| Inflation | 2559 | 4.752 | 2.858 | −0.900 | 15.177 |

*4.2. Regression Results*

After conducting the preliminary diagnostic checks, we proceeded to the regression analysis. At first, we tested whether the extent of carbon emissions differs between financial and non-financial firms. The results reported in Table 2 indicate that the financial dummy had a significant negative impact on total carbon emission ($\beta = -0.845$). We found similar significant negative results for the financial dummies in our robust models that replace total carbon emissions with direct ($\beta = -1.752$) and indirect ($\beta = -0.385$) carbon emissions. Therefore, we confirm that financial firms emit less carbon than non-financial firms. Although this information is a common consensus, we proceeded with this analysis because our main concern in this study is to investigate if the consequences of carbon emissions differ between financial and non-financial firms.

**Table 2.** Do financial firms emit less carbon?

| | Baseline Model | | Robust Model | | | |
|---|---|---|---|---|---|---|
| | Total Carbon Emission | | Direct Carbon Emission | | Indirect Carbon Emission | |
| | OLS | GMM | OLS | GMM | OLS | GMM |
| Carbon emission$_{t-1}$ | | 0.864 *** | | 0.941 *** | | 0.839 *** |
| | | (15.981) | | (17.478) | | (11.925) |
| Financial dummy | −0.845 *** | −0.111 * | −1.752 *** | −0.077 | −0.385 ** | −0.006 |
| | (5.851) | (1.761) | (10.272) | (−0.806) | (3.065) | (0.112) |
| Firm size | 0.415 *** | 0.054 | 0.622 *** | 0.037 | 0.226 * | 0.005 |
| | (5.308) | (1.536) | (6.298) | (0.963) | (2.610) | (0.151) |
| Firm age | 0.031 | −0.006 | 0.082 * | −0.013 | 0.045 | 0.004 |
| | (0.107) | (0.532) | (2.032) | (1.208) | (1.477) | (0.273) |
| Leverage | −0.036 *** | −0.004 | −0.051 *** | −0.003 | −0.017 * | −0.006 |
| | (4.419) | (1.203) | (4.054) | (0.694) | (2.102) | (1.151) |
| Capital expenditure | 0.563 *** | 0.076 * | 0.635 *** | 0.043 | 0.436 *** | 0.094 * |
| | (11.116) | (2.342) | (10.103) | (1.128) | (6.506) | (2.015) |
| Board size | 0.022 *** | 0.002 | 0.031 *** | 0.000 | 0.012 * | 0.006 |
| | (4.284) | (1.031) | (3.625) | (0.147) | (2.086) | (1.391) |
| Independent director on board | −0.002 | 0.000 | −0.002 | 0.000 | 0.002 | 0.000 |
| | (1.429) | (0.445) | (0.899) | (0.138) | (1.718) | (0.715) |
| Strategic ownership | −0.003 | −0.004 | −0.019 | −0.003 | 0.018 | −0.003 |
| | (0.287) | (0.989) | (1.103) | (0.945) | (1.718) | (0.635) |
| GDP growth | −0.007 | 0.000 | −0.007 | 0.000 | −0.011 | 0.002 |
| | (0.607) | (0.124) | (0.425) | (0.154) | (1.171) | (0.609) |
| Inflation | 0.013 | −0.003 | 0.021 * | −0.001 | 0.001 | −0.001 |
| | (1.765) | (1.483) | (2.201) | (0.312) | (0.145) | (0.383) |
| Country fixed effect | Yes | − | Yes | − | Yes | − |
| Year fixed effect | Yes | − | Yes | − | Yes | − |
| Constant | −5.213 *** | −0.253 | −8.136 *** | −0.264 | −3.003 *** | −0.019 |
| | (10.843) | (1.263) | (12.985) | (0.761) | (6.315) | (0.104) |
| Observations | 1496 | 1269 | 1496 | 1269 | 1496 | 1269 |
| F-value/Wald Chi$^2$ | 117.63 *** | 899.45 *** | 93.68 *** | 2118.0 *** | 55.71 *** | 232.43 *** |
| R-square | 0.457 | | 0.446 | | 0.403 | |

Note: We began our analysis by comparing carbon emissions between financial and non-financial firms. We performed this analysis using the following ordinary least squares (OLS) regression model: $Carbon\ emission_{it} = \beta_0 + B_1 Financial\ dummy_{it} + B_2 Controls_{it} + \varepsilon_{it}$. Additionally, we performed a generalized method of moment regression using the following model: $Carbon\ emission_{it} = \beta_0 + B_1 Carbon\ emission_{it-1} + B_2 Financial\ dummy_{it} + B_3 Controls_{it} + \varepsilon_{it}$. Carbon emission is the dependent variable. In our baseline model, we included total carbon emission as the dependent variable. However, we also checked the robustness of our analysis by introducing both direct and indirect carbon emissions as dependent variables in the regression model. The financial dummy takes a score of 1 for financial firms and 0 for non-financial firms. We used the same firm and country-level controls for all regression models. Brief descriptions for all variables included in the regression model are available in Table A2 (Appendix A). ***, **, and * indicate significance at 1, 5, and 10 percent levels, respectively.

We ensured the robustness of our findings reported in Table 2 by conducting a sub-sample analysis and present our findings in Table 3. We found that financial firms emit less total carbon in East Asia ($\beta = -0.652$), Europe ($\beta = -2.513$), Latin America ($\beta = -1.496$), Middle East ($\beta = -4.096$), and South Asia ($\beta = -0.770$). However, we did not find the coefficient of the financial dummy significant for the African sub-sample. While past studies on carbon emissions have explored possible determinants [49] and the expected impact on financial performance [50,51], mostly from the context of the non-financial sector [40] in both advanced and emerging economies, we are the first to empirically estimate that carbon emissions are lower among financial firms compared to non-financial firms in emerging economies. Our findings support the common perception that financial firms are lower emitters than their non-financial peers. The low level of carbon emissions among financial firms in emerging markets provides further justification for the numerous problems identified in past studies in the development of a low-carbon financial sector in emerging markets [52].

**Table 3.** Do financial firms emit less carbon across various regions?

| | East Asia | Europe | Latin America | Middle East | South Asia | Africa |
|---|---|---|---|---|---|---|
| Financial dummy | −0.652 * | −2.513 *** | −1.496 *** | −4.096 *** | −0.770 ** | −0.207 |
| | (2.197) | (8.489) | (7.479) | (6.519) | (5.117) | (1.413) |
| Firm size | 0.421 ** | 1.474 *** | 0.614 *** | 1.741 *** | 0.444 ** | −0.106 |
| | (3.542) | (5.577) | (5.478) | (5.536) | (5.173) | (1.360) |
| Firm age | −0.023 | −0.249 *** | 0.099 * | 0.231 | 0.096 | 0.111 ** |
| | (0.550) | (4.686) | (2.276) | (0.972) | (1.504) | (4.522) |
| Leverage | −0.090 *** | 0.015 | −0.028 * | 0.134 * | −0.030 ** | −0.046 * |
| | (5.196) | (0.798) | (2.188) | (2.221) | (3.612) | (3.135) |
| Capital expenditure | 0.639 *** | 0.025 | 0.515 *** | −0.195 | 0.510 ** | 0.676 *** |
| | (7.083) | (0.204) | (5.315) | (1.139) | (3.940) | (7.847) |
| Board size | 0.018 | 0.009 | 0.007 | 0.069 * | 0.035 ** | 0.035 ** |
| | (1.591) | (0.956) | (0.727) | (2.249) | (4.015) | (3.796) |
| Independent director on board | −0.002 | 0.009 * | −0.001 | 0.004 | −0.014 *** | 0.001 |
| | (0.637) | (2.221) | (0.648) | (0.851) | (6.802) | (0.283) |
| Strategic ownership | 0.052 | −0.190 | −0.013 | −0.077 | −0.116 *** | 0.010 |
| | (1.679) | (2.102) | (1.308) | (0.784) | (11.060) | (0.908) |
| GDP growth | −0.034 * | 0.011 | −0.058* | 0.010 | −0.021 | 0.001 |
| | (2.050) | (0.896) | (2.161) | (0.395) | (1.469) | (0.048) |
| Inflation | 0.013 | 0.027 | −0.081* | 0.045 | 0.007 | 0.013 |
| | (0.467) | (1.708) | (2.721) | (1.285) | (0.639) | (0.747) |
| Constant | −4.557 *** | −5.122 ** | −4.305 *** | −10.038 ** | −0.269 | 0.275 |
| | (6.602) | (4.047) | (5.951) | (3.155) | (0.255) | (0.484) |
| Observations | 471 | 86 | 337 | 39 | 260 | 303 |
| R-square | 0.435 | 0.433 | 0.456 | 0.880 | 0.384 | 0.573 |

Note: We continued with a sub-sample analysis and present the results in Table 3. We split the sample across various regions and performed ordinary least squares (OLS) regression based on the following model: $Carbon\ emission_{it} = \beta_0 + B_1 Financial\ dummy_{it} + B_2 Controls_{it} + \varepsilon_{it}$. In our baseline model, we included total carbon emission as the dependent variable. However, we also checked the robustness of our analysis by introducing both direct and indirect carbon emissions as the dependent variable in the regression model. The financial dummy takes a score of 1 for financial firms and 0 for non-financial firms. We used the same firm and country-level controls for all regression models. Brief descriptions for all variables included in the regression model are available in Table A2 (Appendix A). ***, **, and * indicate significance at 1, 5, and 10 percent levels, respectively.

We proceeded to the second phase of our regression by exploring the determinants of carbon emissions among financial firms in emerging markets. As we have already established earlier that financial firms emit less carbon than non-financial firms, we proceeded with the sample of financial firms for our analysis in Table 4, which presents our results for three determinants of carbon emissions. The first regression model explored the level of carbon emissions among state-owned firms. We found that the coefficient for a state-owned firm dummy was negative ($\beta = -0.071$), indicating that state-owned financial firms in emerging markets emit less carbon. Our results conform to the findings of Tan, Gao, and Komal [53] in that state-owned firms have a higher level of internal control, which leads to a lower level of carbon emissions.

In the final stage of our empirical analysis, we proceeded to investigate our primary research question: Do carbon emissions affect the accounting performance of financial firms differently than non-financial firms? The rationale for such analysis is inherent in the discussion of Jiguang et al. [52] with regards to the slow development of a comprehensive carbon finance market for emerging economies. The financial sector is not actively involved in the development of the carbon finance market [54], which could be attributed to the financing risk and opportunism associated with a carbon-constrained society in emerging economies. As a low-carbon-emitting industry, financial firms may lack the expertise to properly evaluate the risk associated with financing a low-carbon project. Therefore, we aim to raise awareness among financial firms and policymakers by exploring the negative consequences of carbon emission, irrespective of the extent of emissions and the type of firm.

**Table 4.** What affects carbon emission among financial firms?

| | Carbon Emissions | | |
|---|---|---|---|
| State-owned firms | −0.071 *** | | |
| | (−3.61) | | |
| Rent-seeking | | 0.487 ** | |
| | | (2.580) | |
| Environmental project financing | | | −0.295 * |
| | | | (−1.46) |
| Firm size | 0.228 | −0.013 | 0.101 |
| | (1.460) | (−0.08) | (0.590) |
| Firm age | −0.159 | 0.015 | −0.009 |
| | (−1.51) | (0.150) | (−0.09) |
| Leverage | −0.054 * | −0.043 * | −0.024 |
| | (−2.62) | (−2.05) | (−1.06) |
| Capital expenditure | 0.395 *** | 0.422 *** | 0.399 *** |
| | (3.590) | (3.890) | (3.580) |
| Board size | 0.051 * | 0.041 * | 0.048 * |
| | (2.560) | (2.150) | (2.480) |
| Independent director on board | −0.001 | 0.001 | 0.001 |
| | (−0.13) | (0.160) | (0.150) |
| Strategic ownership | 0.010 | 0.023 | 0.029 |
| | (0.400) | (0.930) | (1.100) |
| GDP growth | 0.100 *** | 0.077 ** | 0.074 ** |
| | (3.810) | (2.990) | (2.830) |
| Inflation | −0.005 | 0.021 | 0.023 |
| | (−0.25) | (1.050) | (1.160) |
| Country*Year | Yes | Yes | Yes |
| Constant | 2.554 | 0.059 | −1.030 |
| | (1.610) | (−0.05) | (−0.69) |
| Observations | 110 | 120 | 120 |
| R-square | 0.398 | 0.320 | 0.292 |

Note: We explored possible antecedents of carbon emissions among financial firms using the following models:$Carbon\ emission_{it} = \beta_0 + B_1 State\ owned\ firms_{it} + B_2 Controls_{it} + \varepsilon_{it}$, $Carbon\ emission_{it} = \beta_0 + B_1 Rent - seeking_{it} + B_2 Controls_{it} + \varepsilon_{it}$, $Carbon\ emission_{it} = \beta_0 + B_1 Environmental\ project\ financing_{it} + B_2 Controls_{it} + \varepsilon_{it}$. In all models, we included total carbon emission as the dependent variable. We performed regression of financial firms only. The state-owned firm was a dummy variable and took a value of 1 for state-owned financial firms, and 0 otherwise. We used general and administrative expenses as a proxy for rent-seeking. Environmental project financing was a dummy variable and took a score of 1 if the financial firm financed an environmental project, and 0 otherwise. We used the same firm and country-level controls for all regression models. Brief descriptions for all variables included in the regression model are available in Table A2 (Appendix A). ***, **, and * indicate significance at 1, 5, and 10 percent levels, respectively.

We began our analysis with the impact of carbon emissions on accounting performance. Table 5 presents the results for both financial and non-financial firms. The impact of carbon emissions was negative and significant on the return on assets, our proxy of accounting performance. This negative impact was consistent for both financial and non-financial firms, indicating that the negative impact of carbon emissions on firms' performance does not differ between financial and non-financial firms in emerging markets. We performed an instrumental variable test to control for endogeneity issues in our regression analysis and found similar results. As such, we confirm the robustness of our findings, which are in line with those of Gallego-Álvarez, Segura, and Martínez-Ferrero [55].

We then examined the effect of carbon emission on Tobin's Q, the market-based performance of financial and non-financial firms and the results are presented in Table 6. We found that total carbon emission had a significant negative ($\beta$ = −0.133) impact on the market value, measured by Tobin's Q, of the selected firms. The OLS regression results also hold when we ran 2SLS regression to control for endogeneity issues and confirm the earlier findings of Matsumura, Prakash, and Vera-Munoz [18]. Such results provide a clear indication for the financial sector that carbon emissions are negatively perceived by

stakeholedrs. Therefore, the financial sector in emerging markets needs to address carbon emission issues to ensure market confidence.

**Table 5.** Do carbon emissions affect the accounting performance of financial firms differently than non-financial firms?

| | Return on Assets | | | |
| | OLS | | 2SLS | |
| | Financial | Non-Financial | Financial | Non-Financial |
|---|---|---|---|---|
| Total carbon emission | −0.013 * | −0.013 *** | −0.042 *** | −0.022 *** |
| | (1.872) | (5.171) | (5.661) | (10.726) |
| Firm size | −0.054 *** | −0.017 * | −0.054 *** | −0.013 |
| | (3.817) | (2.208) | (3.957) | (1.721) |
| Firm age | 0.005 | −0.004 | 0.005 | −0.003 |
| | (1.321) | (1.663) | (0.928) | (1.385) |
| Leverage | 0.000 | −0.005 *** | −0.001 | −0.006 *** |
| | (0.100) | (4.867) | (0.537) | (5.298) |
| Capital expenditure | 0.023 | 0.013 ** | 0.035 ** | 0.018 *** |
| | (1.733) | (2.725) | (2.879) | (3.533) |
| Board size | 0.003 | 0.000 | 0.004 ** | 0.000 |
| | (1.762) | (0.855) | (2.691) | (0.548) |
| Independent director on board | 0.000 | 0.000 | 0.000 | 0.000 |
| | (1.082) | (0.809) | (1.067) | (0.735) |
| Strategic ownership | 0.002 | 0.000 | 0.002 | 0.000 |
| | (1.102) | (0.297) | (1.661) | (0.146) |
| GDP growth | 0.002 | 0.005 *** | 0.004 | 0.005 *** |
| | (0.993) | (6.627) | (1.810) | (6.548) |
| Inflation | 0.001 | 0.001 | 0.001 | 0.001 |
| | (0.879) | (1.406) | (1.105) | (1.232) |
| Country*Year | Yes | Yes | Yes | Yes |
| Constant | 0.418 *** | 0.191 *** | 0.417 *** | 0.158 *** |
| | (4.085) | (4.726) | (3.712) | (3.761) |
| Observations | 120 | 1375 | 120 | 1375 |
| R-square | 0.412 | 0.119 | 0.273 | 0.108 |

Note: We explored the impact of carbon emissions on firms' performance and present our findings in Table 5. We performed ordinary least squares (OLS) regression based on the following model: $Financial\ perfomance_{it} = \beta_0 + B_1 Carbon\ emission_{it} + B_2 Controls_{it} + \varepsilon_{it}$. We included return on assets as the dependent variable, which served as a proxy for accounting performance. We tested the robustness of our study findings by conducting the sample analysis with instrumental variables. Total carbon emission was our explanatory variable in these regressions. We used the same firm and country-level controls for all regression models. Brief descriptions for all variables included in the regression model are available in Table A2 (Appendix A). ***, **, and * indicate significance at 1, 5, and 10 percent levels, respectively.

In addition to accounting and market-based performance, we performed an analysis to explore the impact of total carbon emissions on the stability of firms in emerging markets. We report the results in Table 7. Like our earlier results, we found that an increase in carbon emission reduced financial stability, measured by Z-score, for both financial ($\beta = −0.445$) and non-financial firms ($\beta = −0.315$). Safi et al. [56] report a negative association between carbon emissions and financial stability for 7-emerging countries. Therefore, our findings support the generalizability of the earlier findings that an increase in carbon emissions from both financial and non-financial firms leads to an increase in financial instability.

Following Zhang and Tao [57], we explored the impact of carbon emissions on firms' performance using propensity score matching (PSM) technique. Application of the PSM technique allowed us to control for common support problems inherent in regression analysis. We used financial firms as the treatment group and non-financial firms as the control group, performed all regressions on propensity score matching (PSM) samples, and found consistent results, providing further validity of our empirical results. The PSM regression results are exhibited in Table 8. We found results consistent with those of our earlier findings.

**Table 6.** Do carbon emissions affect the market performance of financial firms differently than non-financial firms?

| | Tobin's Q | | | |
| --- | --- | --- | --- | --- |
| | OLS | | 2SLS | |
| | Financial | Non-Financial | Financial | Non-Financial |
| Total carbon emission | −0.133 * | −0.274 *** | −0.305 * | −0.478 *** |
| | (2.275) | (5.286) | (1.812) | (10.640) |
| Firm size | −0.265 | −1.478 *** | −0.264 * | −1.378 *** |
| | (1.887) | (7.280) | (1.999) | (6.815) |
| Firm age | 0.075 | 0.139 ** | 0.070 | 0.163 *** |
| | (1.025) | (2.946) | (1.000) | (3.517) |
| Leverage | −0.009 | −0.003 | −0.015 | −0.009 |
| | (0.657) | (0.113) | (1.216) | (0.354) |
| Capital expenditure | 0.183 ** | 0.594 *** | 0.256 * | 0.707 *** |
| | (2.746) | (7.225) | (2.325) | (7.432) |
| Board size | 0.007 | 0.022 | 0.016 | 0.025 |
| | (0.639) | (1.701) | (1.061) | (1.909) |
| Independent director on board | 0.002 | 0.005 | 0.002 | 0.005 |
| | (0.982) | (1.706) | (1.066) | (1.785) |
| Strategic ownership | −0.027 | 0.054 | −0.024 | 0.049 |
| | (1.451) | (1.826) | (1.223) | (1.673) |
| GDP growth | 0.025 | 0.097 *** | 0.038 * | 0.097 *** |
| | (1.637) | (4.579) | (2.225) | (4.513) |
| Inflation | −0.004 | −0.009 | −0.001 | −0.012 |
| | (0.504) | (0.417) | (0.072) | (0.538) |
| Country * Year | Yes | Yes | Yes | Yes |
| Constant | 3.419 *** | 11.022 *** | 3.411 *** | 10.250 *** |
| | (3.824) | (8.286) | (3.928) | (7.461) |
| Observations | 117 | 1313 | 117 | 1313 |
| R-square | 0.298 | 0.189 | 0.241 | 0.179 |

Note: We explored the impact of carbon emissions on firms' performance and present our findings in Table 6. We performed ordinary least squares (OLS) regression based on the following model: *Financial perfomance$_{it}$ = $\beta_0$ + $B_1$Carbon emission$_{it}$ + $B_2$Controls$_{it}$ + $\varepsilon_{it}$*. We included Tobin's Q as the dependent variable, which served as a proxy for market performance. We tested the robustness of our study findings by conducting the sample analysis with instrumental variables. Total carbon emission was our explanatory variable in these regressions. We used the same firm- and country-level controls for all regression models. Brief descriptions for all variables included in the regression model are available in Table A2 (Appendix A). ***, **, and * indicate significance at 1, 5, and 10 percent levels, respectively.

**Table 7.** Do carbon emissions affect the stability of financial firms differently than non-financial firms?

| | Z-score | | | |
| --- | --- | --- | --- | --- |
| | OLS | | 2SLS | |
| | Financial | Non-Financial | Financial | Non-Financial |
| Total carbon emission | −0.445 ** | −0.315 * | −1.993 | −0.962 *** |
| | (3.261) | (2.478) | (1.131) | (7.040) |
| Firm size | −0.016 | −2.094 *** | 0.930 | −1.775 *** |
| | (0.057) | (6.986) | (0.768) | (5.865) |
| Firm age | 1.405 *** | 0.241 * | 2.261 * | 0.312 ** |
| | (4.113) | (2.147) | (2.122) | (2.743) |
| Leverage | −0.095 * | −0.272 *** | −0.070 | −0.293 *** |
| | (2.616) | (4.922) | (1.388) | (5.182) |
| Capital expenditure | 0.254 | 0.526 *** | 0.367 | 0.883 *** |
| | (2.030) | (4.296) | (1.254) | (5.594) |
| Board size | 0.024 | 0.056 * | 0.062 | 0.066 ** |
| | (0.535) | (2.558) | (0.982) | (2.856) |

**Table 7.** *Cont.*

| | Z-score | | | |
| --- | --- | --- | --- | --- |
| | OLS | | 2SLS | |
| | Financial | Non-Financial | Financial | Non-Financial |
| Independent director on board | −0.021 * | 0.017 ** | −0.072 | 0.018 ** |
| | (2.269) | (3.081) | (1.145) | (3.173) |
| Strategic ownership | −0.138 * | 0.057 | −0.086 | 0.044 |
| | (2.671) | (1.038) | (1.233) | (0.827) |
| GDP growth | 0.237 | 0.264 *** | 0.330 | 0.263 *** |
| | (1.945) | (5.827) | (1.916) | (5.606) |
| Inflation | 0.231 | 0.040 | 0.200 | 0.031 |
| | (1.836) | (1.332) | (1.246) | (0.983) |
| Country*Year | Yes | Yes | Yes | Yes |
| Constant | 1.986 | 17.779 *** | −3.497 | 15.297 *** |
| | (0.587) | (8.707) | (0.451) | (6.845) |
| Observations | 44 | 1354 | 44 | 1354 |
| R-square | 0.807 | 0.199 | 0.617 | 0.175 |

Note: We explored the impact of carbon emissions on firms' performance and present our findings in Table 7. We performed ordinary least squares (OLS) regression based on the following model: $Financial\ perfomance_{it} = \beta_0 + B_1 Carbon\ emission_{it} + B_2 Controls_{it} + \varepsilon_{it}$. We included Z-score as the dependent variable, which served as a proxy for firm stability. We tested the robustness of our study findings by conducting the sample analysis with instrumental variables. Total carbon emission was our explanatory variable in these regressions. We used the same firm- and country-level controls for all regression models. Brief descriptions for all variables included in the regression model are available in Table A2 (Appendix A). ***, **, and * indicate significance at 1, 5, and 10 percent levels, respectively.

**Table 8.** Propensity score matching regression for financial firms.

| | Return on Assets | Tobin's Q | Z-Score |
| --- | --- | --- | --- |
| Carbon emission | −0.142 * | −0.138 * | −1.225 *** |
| | (−1.77) | (−2.23) | (−4.40) |
| Firm size | −0.063 *** | −0.283 * | 0.446 |
| | (−6.12) | (−2.56) | −0.980 |
| Firm age | 0.006 | 0.071 | 1.373 ** |
| | −0.910 | −1.010 | −3.190 |
| Leverage | 0.001 | −0.005 | −0.235 * |
| | −0.750 | (−0.33) | (−2.71) |
| Capital expenditure | 0.027 *** | 0.193 * | 0.246 |
| | −3.410 | −2.110 | −1.010 |
| Board size | 0.002 * | 0.006 | 0.051 |
| | −2.000 | −0.470 | −0.900 |
| Independent director on board | 0.000 | 0.002 | −0.047 * |
| | (−1.27) | −0.630 | (−2.43) |
| Strategic ownership | 0.002 | −0.030 | −0.179 * |
| | −1.100 | (−1.64) | (−2.68) |
| GDP growth | 0.001 | 0.026 | 0.326 * |
| | −0.420 | −1.430 | −2.670 |
| Inflation | 0.000 | −0.006 | 0.148 |
| | −0.010 | (−0.46) | −1.160 |
| Country*Year | Yes | Yes | Yes |
| Observations | 0.426 *** | 3.614 *** | 1.899 |
| | −4.740 | −3.770 | −0.440 |
| Observations | 112 | 109 | 39 |
| R-square | 0.359 | 0.216 | 0.635 |

Note: We explored the impact of carbon emissions on firms' performance with a propensity-matched sample and present our findings in Table 8. We performed ordinary least squares (OLS) regression based on the following model: $Financial\ perfomance_{it} = \beta_0 + B_1 Carbon\ emission_{it} + B_2 Controls_{it} + \varepsilon_{it}$. We included return on assets, Tobin's Q, and Z-score as the dependent variables. We tested the robustness of our study findings by conducting the sample analysis with instrumental variables. Total carbon emission was our explanatory variable in these regressions. We used the same firm- and country-level controls for all regression models. Brief descriptions for all variables included in the regression model are available in Table A2 (Appendix A). ***, **, and * indicate significance at 1, 5, and 10 percent levels, respectively.

We further, examined the robustness of our earlier findings by replacing the proxies of firms' performance. We included earnings per share and credit score as dependent variables. We tested the robustness of our study findings by conducting the sample analysis with instrumental variables. Total carbon emission was our explanatory variable in these regressions. We used the same firm- and country-level controls for all regression models. The results are presented in Table 9. The negative effects of total carbon emission on financial and non-financial firms remain unchanged. In particular, we report that total carbon emissions hurt earnings per share ($\beta = -0.051$) and credit score ($\beta = -2.276$) for financial firms. We report similar findings for non-financial firms. This means that our earlier results are not due to estimation bias or selection of specific variables

**Table 9.** Robust regression results.

| | Earnings Per Share | | | | Credit Score | | | |
|---|---|---|---|---|---|---|---|---|
| | OLS | | 2SLS | | OLS | | 2SLS | |
| | Financial | Non-Financial | Financial | Non-Financial | Financial | Non-Financial | Financial | Non-Financial |
| Total carbon emission | −0.051 * | −0.012 * | −0.115 ** | −0.019 ** | −2.276 *** | −0.320 ** | −1.869 | −0.840 *** |
| | (2.137) | (2.290) | (3.149) | (3.144) | (5.521) | (2.842) | (1.418) | (4.505) |
| Firm size | −0.055 | 0.071 *** | −0.054 | 0.056 *** | 0.259 | 0.538 | 0.263 | 0.284 |
| | (0.699) | (6.798) | (0.725) | (5.258) | (0.275) | (1.871) | (0.297 | (0.944) |
| Firm age | 0.046 | −0.014 * | 0.045 | −0.018* | −0.459 | −0.090 | −0.470) | −0.137 |
| | (1.774) | (2.144) | (1.694) | (2.461) | (0.932) | (0.669) | (0.979) | (1.007) |
| Leverage | 0.002 | −0.008 *** | 0.000 | −0.007 *** | 0.096 | 0.384 *** | 0.081 | 0.401 *** |
| | (0.175) | (3.842) | (0.027) | (3.511) | (0.769) | (9.573) | (0.641) | (10.081) |
| Capital expenditure | 0.080 | −0.025 ** | 0.107 | −0.042 *** | −0.927 | −0.185 | −0.757 | −0.476 * |
| | (1.155) | (2.808) | (1.543) | (4.263) | (1.383) | (1.039) | (0.822) | (2.331) |
| Board size | 0.007 | −0.002 | 0.010 | −0.002 | −0.168 ** | −0.013 | −0.149 * | −0.020 |
| | (1.298) | (1.347) | (1.695) | (1.690) | (3.379) | (0.528) | (2.175) | (0.852) |
| Independent director on board | −0.002 | 0.000 | −0.002 | 0.000 | 0.022 | −0.007 | 0.022 | −0.008 |
| | (0.902) | (1.211) | (0.940) | (1.286) | (1.226) | (1.154) | (1.291) | (1.292) |
| Strategic ownership | −0.009 | −0.003 | −0.008 | −0.002 | 0.000 | −0.164 ** | 0.007 | −0.154 ** |
| | (0.948) | (0.912) | (0.926) | (0.705) | (0.004) | (3.363) | (0.096) | (3.219) |
| GDP growth | 0.015 | 0.006 *** | 0.020 | 0.006 *** | −0.116 | −0.311 *** | −0.086 | −0.311 *** |
| | (1.371) | (4.169) | (1.804) | (3.873) | (1.053) | (6.367) | (0.553) | (6.350) |
| Inflation | 0.006 | 0.001 | 0.008 | 0.001 | 0.099 | −0.153 *** | 0.108 | −0.147 *** |
| | (1.425) | (0.790) | (1.795) | (1.073) | (1.037) | (3.867) | (1.132) | (3.719) |
| Country*Year | Yes | Yes | Yes | Yes | Yes | Yes | Yes | Yes |
| Constant | 0.155 | −0.277 *** | 0.152 | −0.158 ** | 3.309 | 7.476 *** | 3.294 | 9.470 *** |
| | (0.290) | (4.527) | (0.284) | (2.615) | (0.436) | (3.733) | (0.461) | (4.049) |
| Observations | 120 | 1362 | 120 | 1362 | 120 | 1376 | 120 | 1375 |
| R-square | 0.133 | 0.045 | 0.021 | 0.175 | 0.400 | 0.229 | 0.394 | 0.213 |

Note: We performed ordinary least squares (OLS) regression based on the following model: $Financial\ perfomance_{it} = \beta_0 + B_1 Carbon\ emission_{it} + B_2 Controls_{it} + \varepsilon_{it}$. A brief description of all variables included in the regression model is available in Table A2 (Appendix A). ***, **, and * indicate significance at 1, 5, and 10 percent levels, respectively.

Finally, we explored whether past carbon emissions have any impact on future performance. The results, presented in Table 10, indicate that past emissions affect all levels of performance for non-financial firms. However, past emissions only affect market performance and stability for financial firms but not accounting performance. Such findings have wider implications for the financial sector in emerging markets with regard to better integration with public policy to achieve sustainability through ensuring carbon finance for prospective projects.

**Table 10.** Do the past year's carbon emissions affect future performance?

| | Return on Assets | | Tobin's Q | | Credit Score | |
|---|---|---|---|---|---|---|
| | **Financial** | **Non-Financial** | **Financial** | **Non-Financial** | **Financial** | **Non-Financial** |
| Total carbon emission$_{t-1}$ | −0.006 | −0.013 *** | −0.155 * | −0.251 *** | −2.332 *** | −0.345 ** |
| | (1.062) | (4.827) | (2.195) | (4.930) | (4.457) | (2.932) |
| Firm size | −0.046 *** | −0.009 | −0.253 | −1.305 *** | 0.117 | 0.232 |
| | (3.583) | (1.101) | (1.682) | (6.106) | (0.109) | (0.770) |
| Firm age | 0.004 | −0.005 | 0.047 | 0.115* | −0.344 | −0.055 |
| | (0.739) | (1.583) | (0.590) | (2.231) | (0.624) | (0.341) |
| Leverage | 0.000 | −0.005 *** | −0.009 | 0.003 | 0.098 | 0.386 *** |
| | (0.310) | (3.709) | (0.635) | (0.095) | (0.802) | (7.769) |
| Capital expenditure | 0.021 | 0.008 | 0.213 ** | 0.501 *** | −0.910 | 0.014 |
| | (1.610) | (1.512) | (2.780) | (5.217) | (1.159) | (0.069) |
| Board size | 0.001 | −0.001 | 0.005 | 0.016 | −0.180 ** | −0.026 |
| | (1.032) | (1.401) | (0.338) | (1.224) | (3.145) | (1.050) |
| Independent director on board | −0.001 | 0.000 | 0.001 | 0.005 | 0.020 | −0.008 |
| | (1.578) | (0.670) | (0.603) | (1.411) | (0.964) | (1.163) |
| Strategic ownership | 0.000 | 0.001 | −0.025 | 0.056 | 0.007 | −0.179 ** |
| | (0.330) | (0.636) | (1.310) | (1.699) | (0.071) | (3.229) |
| GDP growth | 0.001 | 0.005 *** | 0.012 | 0.102 *** | −0.049 | −0.331 *** |
| | (0.494) | (7.299) | (0.655) | (4.714) | (0.405) | (6.897) |
| Inflation | 0.000 | 0.001 * | −0.007 | −0.009 | 0.089 | −0.160 *** |
| | (0.184) | (2.125) | (0.853) | (0.370) | (0.916) | (3.743) |
| Country*Year | Yes | Yes | Yes | Yes | Yes | Yes |
| Constant | 0.379 *** | 0.142 *** | 3.351 ** | 10.010 *** | 4.195 | 9.104 *** |
| | (3.965) | (3.613) | (3.417) | (7.619) | (0.489) | (4.403) |
| Observations | 104 | 1165 | 102 | 1130 | 104 | 1165 |
| R-square | 0.390 | 0.089 | 0.285 | 0.151 | 0.256 | 0.239 |

Note: We explored the lag effect of carbon emissions on firms' performance and present our findings in Table 10 We performed ordinary least squares (OLS) regression based on the following model: *Financial perfomance$_{it}$* = $\beta_0$ + $B_1$*Carbon emission$_{it}$* + $B_2$*Controls$_{it}$* + $\varepsilon_{it}$. We included return on assets, Tobin's Q, and Z-score as the dependent variables. We tested the robustness of our study findings by conducting the sample analysis with instrumental variables. Total carbon emission was our explanatory variable in these regressions. We used the same firm- and country-level controls for all regression models. Brief descriptions for all variables included in the regression model are available in Table A1 (Appendix A). ***, **, and * indicate significance at 1, 5, and 10 percent levels, respectively.

## 5. Conclusions

The impact of firms' carbon emissions on performance has gained renewed interest from academia and policymakers, partly due to the rise in increased shareholder activism but mainly due to the unprecedented upsurge of carbon stock in the atmosphere. The increasing trend of anthropogenic emissions owing mainly to the rising activities of corporate firms and the resulting adverse economic and social consequences on human ecology have prompted stakeholders to scrutinize firms' carbon emissions and their impact on the firms' financial health. In this pursuit, this paper has attempted to examine the effects of firms' direct and indirect carbon emissions on their accounting and market-based performance. In so doing, we collected data from 22 emerging economies across six continents. An unbalanced panel of 2591 observations for the period 2011–2020 was analyzed applying the OLS and 2SLS regression methods. To dilute the firm- and country-level effects, we included firm-specific and macroeconomic variables in the model. In addition, the robustness of the results was checked using propensity score matching and applying alternative proxies for performance.

The results reveal that non-financial firms emit more carbon than their financial peers. The difference is more pronounced for direct carbon emissions than the indirect carbon emissions and for Middle Eastern countries compared to other regions. However, carbon emissions reduce return on assets and Tobin's Q for both types of firms, although the magnitude of the effect is higher for non-financial firms than their financial counterparts. Similarly, the effect of carbon emissions on stability is negative for both types of entities, but the stability of financial firms is affected more by carbon emissions than non-financial

firms. Our results remain valid even if we apply alternative proxies for performance and estimation techniques.

These findings offer several policy implications for policymakers, managers, and investors. First, in most cases, financial firms are not equally disciplined compared to their non-financial peers. This conclusion is derived from the fact that the magnitude of negative impact of carbon emissions is more pronounced for non-financial firms than their financial counterparts. This can mainly be attributed to the lack of awareness of people regarding the carbon emitted by financial firms, which are considered clean and low polluters. However, our point of view is that emissions are emissions regardless of their sources of origin. Hence, financial firms are to be equally scrutinized for the greater interest of carbon mitigation plans. Emissions from financial firms are both direct and indirect. For the mitigation of direct emissions, financial institutions can consider constructing physical facilities that are energy efficient. For instance, branches can use solar power and equipment can be devised in such a fashion to consume less energy. For tackling indirect emission, financial institutions can finance eco-friendly firms at a discounted rate. Regulatory authorities can formulate suitable policies to achieve this objective. For example, a portion of the national climate fund can subsidize loans that are aimed at environmental projects and green firms. Moreover, carbon emissions are positively associated with financial instability. Hence, regulators can support financial stability by providing more incentives for investment in low-carbon-emitting technologies.

Second, firms' managers should consider carbon mitigation strategies seriously because carbon emissions negatively affect shareholder value. This means that managers can enhance shareholders' value by undertaking emission abatement policies to boost their financial and market performance. Managers are expected to be aware of their company's carbon emission strategies to protect them from a decline in profitability. They can strategize long-, medium-, and short-term carbon reduction targets commensurate with national and international goals. Moreover, carbon-cutting strategies help firms increase their credit score and financial stability, which may facilitate boosting investors' confidence in low-emitting firms. This may provide opportunities for firms to fund them at a lower cost.

Third, our findings have important implications for investors. Firms profiled as having high carbon emissions receive lower credit ratings, which indicates an elevated default probability. Hence, investors are expected to follow the required precautions before investing in such firms. Moreover, the market value and profitability of firms decline with the increase in carbon emissions. As a result, investors should take firms' carbon emissions into their investment decisions. Our results also confirm that rent-seeking firms, firms that have higher general and administrative expenses, have higher emissions. We have argued that firms with higher administrative expenses may suffer from a shortage of resources required for adopting carbon abatement technologies. Hence, investors should carefully consider such firms as a place for their valuable investments.

Our research examines if the effects of carbon emissions differ between financial and non-financial firms in emerging economies. Future research may endeavor to study if these results hold for developed and developing countries. Moreover, investigating the effects of emissions in carbon-intensive and carbon-non-intensive firms would be worth pursuing in the future to complement our results.

**Author Contributions:** Conceptualization, M.D.M.; methodology, R.H.; software, R.H.; formal analysis, M.D.M. and R.H.; data curation, M.U.; writing—review and editing, M.D.M. and M.U. All authors have read and agreed to the published version of the manuscript.

**Funding:** This research received no external funding.

**Institutional Review Board Statement:** Not applicable.

**Informed Consent Statement:** Not applicable.

**Data Availability Statement:** Available upon request to authors.

**Conflicts of Interest:** The authors declare no conflict of interest.

## Appendix A

**Table A1.** Sample distribution.

| Regions | Financial | | Non-Financial | | Total | |
|---|---|---|---|---|---|---|
| | Firms | Observations | Firms | Observations | Firms | Observations |
| East Asia and Pacific | 37 | 122 | 128 | 500 | 165 | 622 |
| China | 22 | 62 | 41 | 86 | 63 | 148 |
| Indonesia | - | - | 10 | 41 | 10 | 41 |
| Malaysia | 6 | 20 | 30 | 147 | 36 | 167 |
| Philippines | 3 | 9 | 16 | 73 | 19 | 82 |
| Thailand | 6 | 31 | 31 | 153 | 37 | 184 |
| Europe and Central Asia | 10 | 65 | 22 | 108 | 32 | 173 |
| Hungary | 1 | 10 | 2 | 20 | 3 | 30 |
| Turkey | 9 | 54 | 20 | 88 | 29 | 142 |
| Latin America and Caribbean | 25 | 123 | 108 | 606 | 133 | 729 |
| Argentina | 4 | 9 | 11 | 22 | 15 | 31 |
| Brazil | 6 | 56 | 56 | 363 | 62 | 419 |
| Colombia | 6 | 27 | 13 | 71 | 19 | 98 |
| Mexico | 8 | 29 | 23 | 139 | 31 | 168 |
| Peru | 2 | 2 | 5 | 11 | 7 | 13 |
| Middle East and North Africa | 7 | 32 | 12 | 44 | 19 | 76 |
| Kuwait | - | - | 2 | 7 | 2 | 7 |
| Morocco | - | - | 1 | 5 | 1 | 5 |
| Oman | 1 | 6 | - | - | 1 | 6 |
| Qatar | 2 | 11 | - | - | 2 | 11 |
| Saudi Arabia | - | 5 | 8 | 25 | 8 | 30 |
| United Arab Emirates | 3 | 11 | 2 | 7 | 5 | 17 |
| South Asia | 7 | 41 | 55 | 303 | 62 | 344 |
| India | 7 | 41 | 56 | 296 | 63 | 337 |
| Sri Lanka | - | - | 1 | 7 | 1 | 7 |
| Sub-Saharan Africa | 18 | 140 | 65 | 507 | 83 | 647 |
| Kenya | - | - | - | 6 | - | 6 |
| South Africa | 18 | 140 | | 501 | 18 | 641 |
| Total | 104 | 523 | 328 | 2068 | 432 | 2591 |

**Table A2.** Variable definition.

| Variables | Definition |
|---|---|
| Panel A: Performance variables. | |
| Return on assets | Net income prior to financing costs divided by total assets. |
| Earnings per share | Net profit (on continuous activities) divided by the weighted average number of shares outstanding during the period. |
| Tobin's Q | Market value of firm / book value of assets. |
| Credit rating | Agency-equivalent credit rating implied by the current estimated forward 1-year SCR default probability. High score indicates good credit rating and vice versa. |
| Z-score | (Capital asset ratio + Mean Return on Asset) / Standard Deviation of Return on Asset |
| Panel B: Carbon emission proxies. | |
| Total carbon emission | Total carbon dioxide ($CO_2$) and $CO_2$ equivalent emissions in tonnes. |
| Direct carbon emission | Direct emissions of $CO_2$ and $CO_2$ equivalents emissions in tonnes. |
| Indirect carbon emission | Indirect emissions of $CO_2$ and $CO_2$ equivalent emissions in tonnes. |
| Panel C: Exploratory variable. | |
| Financial dummy | Dummy variable. Financial firms take the value of 1, and 0 otherwise. |
| Panel D: Firm controls | |
| Firm size | Natural logarithm of total assets. |

**Table A2.** *Cont*.

| Variables | Definition |
|---|---|
| Firm age | Total number of years the firm is in operation since incorporation. |
| Leverage | Total debt divided by total equity. |
| Capital expenditure | Capital expenditure is the funds used by a company to acquire or upgrade physical assets. |
| Board size | Total number of directors on the corporate board. |
| Independent members on board | Percentage of independent members on the corporate board. |
| Strategic ownership | The number of shares held by strategic investors (corporations, holding companies, individuals, and government agencies). |
| Panel E: Country controls | |
| GDP growth | Annual growth in the Gross Domestic Product (GDP). |
| Inflation | Annual percentage change in the consumer price. |

**Table A3.** Correlation matrix.

| | | 1 | 2 | 3 | 4 | 5 | 6 | 7 | 8 | 9 | 10 | 11 | 12 | 13 | 14 |
|---|---|---|---|---|---|---|---|---|---|---|---|---|---|---|---|
| 1 | Return on assets | 1.000 | | | | | | | | | | | | | |
| 2 | Tobin | 0.646 * | 1.000 | | | | | | | | | | | | |
| 3 | Credit score | −0.428 | −0.469 * | 1.000 | | | | | | | | | | | |
| 4 | Total carbon emission | −0.021 | −0.099 * | 0.020 | 1.000 | | | | | | | | | | |
| 5 | Financial | −0.224 * | −0.228 * | 0.101 * | −0.448 * | 1.000 | | | | | | | | | |
| 6 | Firm size | −0.268 * | −0.326 * | 0.094 * | 0.111 * | 0.537 * | 1.000 | | | | | | | | |
| 7 | Firm age | −0.058 * | 0.012 | 0.017 | 0.145 * | −0.019 | 0.084 * | 1.000 | | | | | | | |
| 8 | Leverage | −0.300 * | −0.189 * | 0.230 * | −0.259 * | 0.623 * | 0.565 * | 0.048 * | 1.000 | | | | | | |
| 9 | Capital expenditure | −0.027 | −0.100 * | −0.014 | 0.571 * | −0.267 * | 0.646 * | 0.037 | 0.013 | 1.000 | | | | | |
| 10 | Board size | −0.069 * | −0.107 * | 0.057 * | 0.170 * | 0.101 * | 0.278 * | 0.098 * | 0.101 * | 0.284 * | 1.000 | | | | |
| 11 | Independent director on board | −0.010 | 0.060 * | 0.037 | 0.005 | −0.044 * | −0.187 * | 0.091 * | −0.075 * | −0.149 * | −0.06 * | 1.000 | | | |
| 12 | Strategic ownership | 0.037 | 0.015 | −0.170 * | 0.156 * | 0.000 | 0.295 * | −0.189 * | 0.051 * | 0.364 * | 0.008 | −0.331 * | 1.000 | | |
| 13 | GDP growth | 0.162 * | 0.160 * | −0.288 * | 0.067 * | 0.034 | 0.168 * | −0.039 | −0.004 | 0.183 * | −0.071 * | −0.110 * | 0.335 * | 1.000 | |
| 14 | Inflation | 0.044 * | −0.001 | −0.052 * | −0.022 | −0.013 | −0.082 * | 0.098 * | 0.024 | 0.007 | −0.004 | −0.0637 * | −0.1894 * | −0.083 * | 1.000 |

* indicate significance at 10 percent level.

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
