# Peer review of "Carbon Emissions and Firm Performance: Evidence from Financial and Non-Financial Firms from Selected Emerging Economies"

_sustainability, doi:10.3390/su132313281_

Round 1
Reviewer 1 Report
This paper deals with interesting and actual topic - it examines the impact of firms' carbon emission on their accounting and market-based performance.
Hypothesis is clearly stated and tested using appropriate sample (104 financial and 432 non-financial firms from 22 emerging economies). Authors process the date using appropriate statistic methods.
Finally, paper provides possible solutions for managers, policymakers and investors to reduce negative impact of carbon emission on firms' profitability.
Author Response
Reviewer comment: This paper deals with interesting and actual topic - it examines the impact of firms' carbon emission on their accounting and market-based performance.
Response: We would like to thank the reviewer for his positive comments.
Reviewer comment: Hypothesis is clearly stated and tested using appropriate sample (104 financial and 432 non-financial firms from 22 emerging economies). Authors process the date using appropriate statistic methods.
Response: We would like to thank the reviewer for his positive comments.
Reviewer comment: Finally, paper provides possible solutions for managers, policymakers and investors to reduce negative impact of carbon emission on firms' profitability.
Response: We would like to thank the reviewer for his positive comments.
Reviewer 2 Report
Thank you for the opportunity of reading and reviewing your interesting manuscript. It addresses a topic of real importance and which is under the journal s scope. The paper is well written, with a good structure and uses adequate research methodology. I have several suggestions for improving the article and its readability:
1.The title is not quite clear: I suggest something like: Carbon Emission and Firm Performance: Evidence from Financial and Non-financial Firms from selected Emerging Economies .
2.In the Introduction section there are issues that would be better placed in the Literature section and in the Methodology section. Please consider moving these parts.
3.some sentences need citation/sources. For example lines #73-76
4.there is only one hypothesis, which can be seen as bringing little contribution to the existing knowledge. However, the main merit resides in the extension of the sample.
5.there are many tables with econometric results, you can consider putting some of them in an appendix
6.given the extreme importance of the findings, I suggest enhancing the final section and retrieving more implications/suggestions for policies to be implemented.
Good luck!
Author Response
We would like to thank the reviewers for their valuable comments which have helped improve the paper. The following is the reviewers’ comments and our response:
Reviewer observation: Thank you for the opportunity of reading and reviewing your interesting manuscript. It addresses a topic of real importance and which is under the journal s scope. The paper is well written, with a good structure and uses adequate research methodology. I have several suggestions for improving the article and its readability:
Response: We would like to thank the reviewer for the positive view on our manuscript.
Reviewer comment 1: The title is not quite clear: I suggest something like: Carbon Emission and Firm Performance: Evidence from Financial and Non-financial Firms from selected Emerging Economies.
Response: We thank the reviewer for this important comment. We have revised the title as per the suggestion.
Reviewer comment 2: In the Introduction section there are issues that would be better placed in the Literature section and in the Methodology section. Please consider moving these parts.
Response: We thank the reviewer for this suggestion. All the indicated paragraphs haven been moved from the introduction to literature review and methodology as they fit.
Reviewer comment 3: some sentences need citation/sources. For example lines #73-76
Response: We thank the reviewer for notifying this issue. Reference is provided and inserted the same in the reference section
Reviewer comment 4: There is only one hypothesis, which can be seen as bringing little contribution to the existing knowledge. However, the main merit resides in the extension of the sample.
Response: We thank the reviewer for this important comment. We concur with the reviewer that our main contribution is extending the hypothesis through analysis.
Reviewer comment 5: there are many tables with econometric results, you can consider putting some of them in an appendix
Response: We thank the reviewer for this valuable comment. We have moved two tables to the Appendix.
Reviewer comment 6: given the extreme importance of the findings, I suggest enhancing the final section and retrieving more implications/suggestions for policies to be implemented.
Response: We thank the reviewer for this this comment. We have significantly extended the implications of the concluding sections adding more policy recommendations.